# Hidden in the Noise: Two-Stage Robust Watermarking for Images

**Kasra Arabi, Benjamin Feuer, R. Teal Witter, Chinmay Hegde, Niv Cohen**
New York University

## Abstract

As the quality of image generators continues to improve, deepfakes become a topic of considerable societal debate. Image watermarking allows responsible model owners to detect and label their AI-generated content, which can mitigate the harm. Yet, current state-of-the-art methods in image watermarking remain vulnerable to forgery and removal attacks. This vulnerability occurs in part because watermarks distort the distribution of generated images, unintentionally revealing information about the watermarking techniques.

In this work, we first demonstrate a distortion-free watermarking method for images, based on a diffusion model's initial noise. However, detecting the watermark requires comparing the initial noise reconstructed for an image to all previously used initial noises. To mitigate these issues, we propose a two-stage watermarking framework for efficient detection. During generation, we augment the initial noise with generated Fourier patterns to embed information about the group of initial noises we used. For detection, we (i) retrieve the relevant group of noises, and (ii) search within the given group for an initial noise that might match our image. This watermarking approach achieves state-of-the-art robustness to forgery and removal against a large battery of attacks.

## 1 Introduction

Generative AI is capable of synthesizing high-quality images indistinguishable from real ones. This capability can be used to deliberately deceive. These fake image generations, called deepfakes, have the potential to cause severe societal harms through the spread of confusion and misinformation (Peebles & Xie, 2022; Esser et al., 2024; Chen et al., 2024; Ramesh et al., 2021). In addition, owners of different models and images may want to control the spread of their derivatives for copyright reasons and safeguard their intellectual property. One way to mitigate these harms is model watermarking. The study of watermarking has a rich history and has recently been adopted for AI-generated content (Pun et al., 1997; Langelaar et al., 2000; Craver et al., 1998). For an extended discussion of recent work in this area, we direct the reader to Appendix B. Unfortunately, most current image watermarking methods are not robust to watermark removal attacks utilizing image diffusion generative models (Zhao et al., 2023a).

Recently, new watermarking methods utilize the inversion property of DDIM to achieve more robust watermarking (Wen et al., 2023; Ci et al., 2024; Yang et al., 2024b). These methods embed patterns in a diffusion model's initial noise and then detect them in the noise pattern reconstructed from the generated image. This technique provides strong robustness against various attacks, making it effective at resisting watermark removal attempts. Yet, these prior methods are themselves vulnerable to new types of attacks. Tree-Ring Wen et al. (2023) add a pattern to the initial noise,

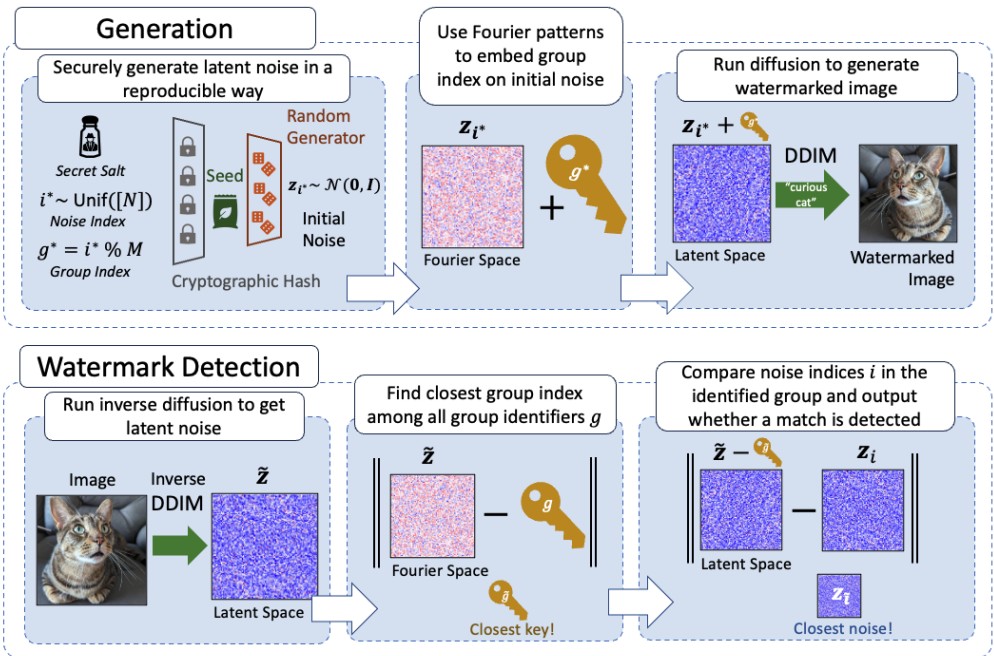

Figure 1: **The WIND method for robust image watermaking**. The method is designed to use $N$ possible initial noises splitted to $M$ groups. **Generation:** Using a secret salt and an index $i^*$, we securely and reproducibly generate initial noise $\mathbf{z}_{i^*}$. We then embed a group index $g^*$ of that noise to make easier retrieval possible and embed it using a Fourier pattern. Finally, we run diffusion with the embedded latent noise to produce a watermarked image. **Detection:** We reconstruct the initial noise $\tilde{\mathbf{z}}$. Next, we search over the possible group indices $g$ for the closest Fourier pattern to the one embedded in $\tilde{\mathbf{z}}$. We then look over initial noises in group $\tilde{g}$ to find the match.

making it distinct from a random Gaussian initial noise in a way that an attacker can detect (Yang et al., 2024a). This may enable forgery attacks, aimed at applying the watermark without the owner's permission. Such attacks are often even more concerning than removal attacks, as they can cause severe damage to model owners if their watermark is associated with illegal content.

Therefore, there is a need for image watermarking methods that generate images that are not distinct from non-watermarked images (to anyone but the model owner). As suggested by previous works, since the model already takes random noise as initialization, we may initialize it with a pseudo-random noise pattern that we can detect later (Yang et al., 2024b; Kuditipudi et al., 2023). Namely, reconstructing an approximation of the *initial noise* used in the diffusion process from a given image allows the detection of the noise pattern used by the model. Although this reconstructed noise is not completely identical to the used *initial noise*, it is much more similar to the initial noise than it is to other randomly distributed noise patterns. Thus, it can serve as a watermark that can be identified in the generated images (see also Appendix C for similar ideas used in previous works).

While using a pseudo-random initial noise does not distort the distribution of single-generated images, it may carry information about the watermark when groups of images are examined together. Specifically, works such as Yang et al. (2024b) embeds the watermark in an initial noise such that

the resulting generated image comes from the same distribution as non-watermarked images. Yet, when many images generated from the same noise pattern are examined together, the correlation between them may expose that they are not distortion-free as a set. E.g., the average of many similarly-watermarked images may differ from the average of non-watermarked ones (Yang et al., 2024a). A natural solution to this distortion of sets is to use more than one initial noise for each watermark we deploy.

Yet, given a sufficiently small set of initial noises (denoted as $N$) and an enormous number of images generated by a model, an attacker could potentially still collect many images sharing the same initial noise in order to perform removal and forgery attacks as was applied to previous methods (Yang et al., 2024a). Using many initial noises (a large value of $N$) will make such attacks much more difficult, if not infeasible. Surprisingly, we find that a very large number of random initial noises remain distinguishable from one another, even after reconstructing the noise from a generated image. However, a large value of $N$ might incur a negative effect on the runtime and accuracy of the approach. In order to lower the effective quantity of noises we need to scan at detection while retaining strong robustness, we propose a two-stage efficient watermarking framework. We supplement our $N$ initial noise samples with $M$ Fourier patterns as a *group identifier* - a unique identifier of a subset of initial noises we might have used for generating a given image (Figure 1). During detection, we may first recover the group identifier (stage 1) and use it to find an exact match (stage 2). Thus, we reduce our search space to the number of initial noises per group ($N/M$).

Our key contributions are as follows:

1. We demonstrate that the initial noise used in the diffusion process is itself a distortion-free watermarking method for images (Section 3).

2. We present WIND, our two-stage method for effectively using the initial noise as a watermark (Section 4).

3. We demonstrate that WIND achieves state-of-the-art results for its robustness to removal and forgery attempts (Section 5).

## 2 Preliminaries

### 2.1 Threat Model

In a watermarking scheme we usually consider the owner, trying to mark images as an output of their model; and an attacker, trying to remove or forge the watermark on unrelated images.

**The Owner** releases a private model (diffusion model in our case) that clients can access through an API, allowing them to generate images that contain a watermark. The watermark is designed to have a negligible impact on the quality of the generated images. There are a few settings regarding the watermark detection, including public infomation and private information watermarking (Cox et al., 2007; Wong & Memon, 2001). We focus on the setting where the watermark is detectable only by the owner, enabling them to verify whether a given image was generated by their model using private information.

**The Attacker** uses the API to generate an image and subsequently attempts to launch a malicious attack aimed at either removing or forging the embedded watermark, with the intention of using the image or watermark for unauthorized purposes.

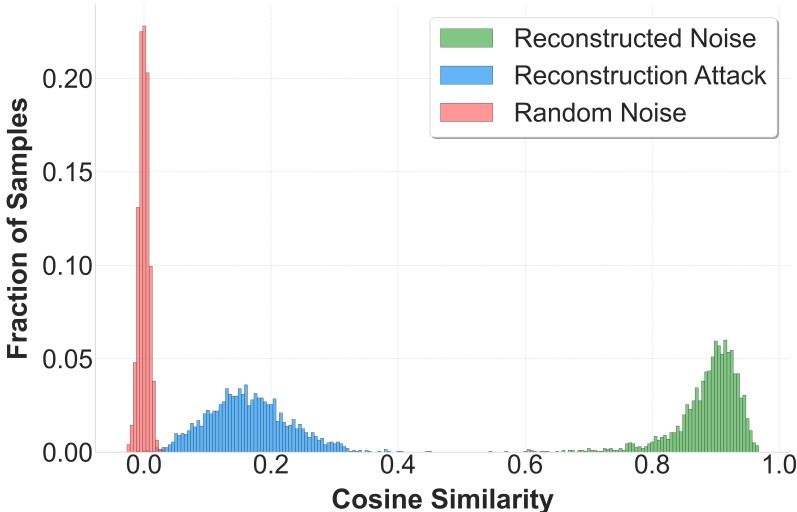

Figure 2: Cosine similarity distribution between initial noise, and: (i) a noise reconstructed from a watermarked image (reconstructed noise) (ii) a noise reconstructed from a forged image using a public model to imitate our watermarked image (reconstruction attack, described in Section 3). (iii) Random noise. These results are reliant on the approximate inversion of DDIM without the ground-truth prompt.

## 2.2 DIFFUSION MODELS INVERSION

Diffusion model inversion aims to find the reconstructed noise representation of a given data point, effectively reversing the generative process. Let $T$ be the number of diffusion steps, in both the generation and inversion processes. In the standard generation process, we start with noise $\tilde{\mathbf{x}}_T$ drawn from an appropriately scaled Gaussian and iteratively apply $\tilde{\mathbf{x}}_t = \tilde{\mathbf{x}}_{t+1} + \epsilon_\Theta(\tilde{\mathbf{x}}_{t+1})$, where $\epsilon_\Theta$ is a trained model that predicts the noise to be removed and $t \in [T]$ is the time step describing how much noise should be removed in each stage. Conversely, the inversion process begins with a data point $\hat{\mathbf{x}}_0$ and moves towards its reconstructed noise representation by applying $\hat{\mathbf{x}}_{t+1} = \hat{\mathbf{x}}_t - \epsilon_\Theta(\hat{\mathbf{x}}_t)$. This process relies on the assumption that $\epsilon_\Theta(\hat{\mathbf{x}}_{t+1}) \approx \epsilon_\Theta(\hat{\mathbf{x}}_t)$, allowing us to approximately invert the diffusion process by adding the predicted noise (Ho et al., 2020; Song et al., 2022). DDIM's efficient sampling allows this technique to be particularly useful (Song et al., 2022).

## 2.3 TREE-RING AND RINGID WATERMARKS

In order to watermark images in a human-imperceptible and robust way, previous works have encoded specific patterns in the Fourier space of the initial noise. *Tree-Ring* (Wen et al., 2023) first transforms the initial noise into the Fourier space. A key pattern is then embedded into the center of the transformed noise. The noise is subsequently transformed back into the spatial domain. During the detection phase, the diffusion process is inverted, and the Fourier domain is examined to verify the presence of the imprinted pattern. *RingID* (Ci et al., 2024) shows that Tree-Ring struggles to distinguish between different keys. Therefore, the number of unique keys (distinguishable from one another) that can be embedded with Tree-Ring is low. They increase the possible number of unique keys that can be encoded using Fourier patterns.

**Systematic Distribution Shifts in Generated Images Enable Attacks.**   Systematic distribution shifts in the generated content make it easier to verify the existence of a watermark. However, in the case of Tree-Ring and other watermarking techniques, it also opens up an avenue of attack (Wen et al., 2023; Yang et al., 2024b; Xian et al., 2024; Bui et al., 2023). Emblematic is the method of Yang et al. (2024a), whose attack approximates the difference between watermarked and non-watermarked images. Increasing the number of images with the watermark can improve the accuracy of the approximation. The impact of distribution shifts is significant, as the attack remains effective even when the watermarked and non-watermarked images are not paired (Figure 3).

---

**Algorithm 1** Generation Algorithm

---

1: **Input:** $N$: number of initial noises, $M$: number of groups, $s$: secret salt, $p$: prompt, $\Theta$: private model weights
2: Sample initial noise index $i^* \sim \text{Unif}([N])$
3: Compute group identifier $g^* = i^* \% M$                                        ▷ Modulus of initial noise index
4: Calculate embedding of the group identifier $g_{emb}(g^*)$
5: Securely generate $\texttt{seed} = \texttt{hash}(i^*, s)$                   ▷ Apply cryptographic hash function
6: Sample $\mathbf{z}_{i^*} \sim \mathcal{N}(\mathbf{0}, \mathbf{I})$ from a pseudorandom generator with $\texttt{seed}$
7: Add the identifier embedding $g_{emb}(g^*)$ to $\mathbf{z}_{i^*}$ to get $\mathbf{z}_{i^*\_emb}$
8: **return** $\texttt{image} = G_\Theta(\mathbf{z}_{i^*\_emb}, p)$                          ▷ Diffusion process $G$ with weights $\Theta$

---

---

**Algorithm 2** Detection Algorithm *(WIND_fast)*

---

1: **Input:** $\texttt{image}$: (possibly) watermarked image, $N$: number of initial noises, $M$: number of groups, $s$: secret salt, $\Theta$: private model weights, $\tau$ : threshold for detection
2: Recover reconstructed noise $\tilde{\mathbf{z}} = G_\Theta^{-1}(\texttt{image})$                  ▷ Inverse diffusion with private weights
3: Extract closest group identifier $\tilde{g}$ from group identifier embedding in $\tilde{\mathbf{z}}$
4: **for** $i \in [N]$ such that $i \% M = \tilde{g}$ **do**                         ▷ Search over subset of initial noise indices
5:     Build initial noise $\mathbf{z}_i$ using secret salt $s$ and $\texttt{hash}$                         ▷ As in Algorithm 1
6:     Compare $\mathbf{z}_i$ to $\tilde{\mathbf{z}}$ after removing Fourier embedding $\tilde{g}$
7: **end for**
8: **if** any noises are closer than threshold $\tau$ **then**
9:     Declare "watermarked"
10: **else**
11:     Declare "not watermarked"
12: **end if**

---

## 3   INITIAL NOISE IS A DISTORTION FREE WATERMARK

Watermarks which systematically perturb the distribution of image generations are more vulnerable to removal and forgery attacks. A distortion-free watermarking method, by contrast, is more robust (Kuditipudi et al., 2023). Our first finding is that the initial noise already in standard use in diffusion models can be such a watermark.

Let $N$ be the number of initial noises we can generate. We will secure our watermarking process with a long, secret salt $s$. We begin by sampling a random (and reproducible) initial noise. Let $i^* \sim \text{Unif}([N])$ be the index of the initial noise. We will use a hash function to get a seed $\text{hash}(i^*, s)$. Plugging the seed into a pseudorandom generator, we generate a reproducible initial noise vector

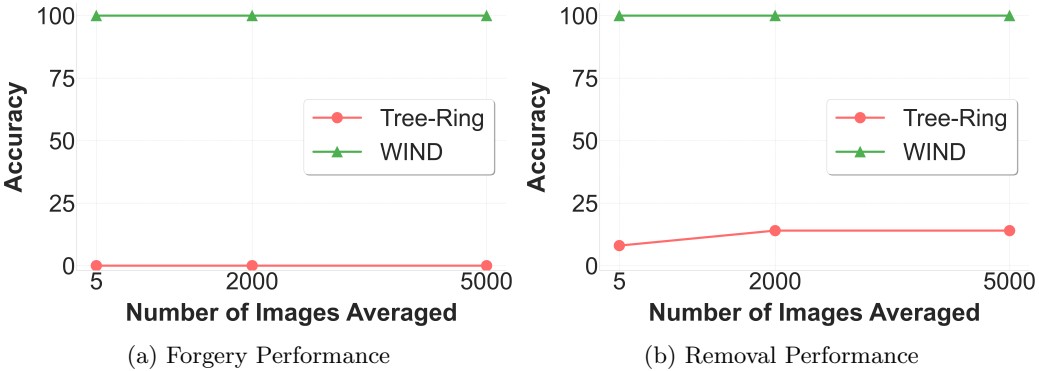

(a) Forgery Performance        (b) Removal Performance

Figure 3: Detection accuracy for forgery and removal attacks using Yang et al. (2024a). A value of 0 represents complete failure (the attacker successfully removed the watermark or forged it onto another image), while 100 indicates perfect defense (no watermark removal or forgery occurred).

$\mathbf{z}_{i^*} \sim \mathcal{N}(\mathbf{0}, \mathbf{I})$ drawn from a centered Gaussian distribution. When we generate fewer than $N$ images, we can use each initial noise at most once and the noise appears distortion-free. We discuss the case when the number of images exceeds $N$ in Appendix F.

**Empirical validation of initial noise watermarking.** To empirically validate our claim that the initial noise can serve as a watermark, we compute the cosine similarity between the initial noise $\mathbf{z}_{i^*}$ and (i) random noise $\mathbf{z} \sim \mathcal{N}(\mathbf{0}, \mathbf{I})$, (ii) the reconstructed noise $\tilde{\mathbf{z}}$ when we have access to the private model weights, and (iii) the reconstructed noise $\tilde{\mathbf{z}}^{\text{attack}}$ from an image imitating our noise pattern without access to the private model weights. The imitation attempt is done by inversing our watermarked image back into noise, and generating a new image from it; where both steps are done using a public model as described in *reconstruction attack* below (we used Stable Diffusion-v2 (Rombach et al., 2022) for the experiment, as it is the most similar model to our watermarking model).

During the watermarking process, we create an image `image` through diffusion with the private model weights $\Theta$ conditioned on a private text prompt $p$. Formally, $\texttt{image} = G_\Theta(\mathbf{z}_{i^*}, p)$. We obtain the reconstructed noise that we use for detection via an inverse diffusion process $G^{-1}$. Formally, $\tilde{\mathbf{z}} = G_\Theta^{-1}(\texttt{image})$.

**Reconstruction Attack.** An attacker trying to forge the watermarked image will not have access to our private weights, instead they will have some other weights $\Theta'$. Using the same starting watermarked image, they will attempt to recover the initial noise. Let $\tilde{\mathbf{z}}' = G_{\Theta'}^{-1}(\texttt{image})$. Then, with this initial noise, they will generate a forged image with (possibly offensive) text prompt $p'$, producing $\texttt{image'} = G_{\Theta'}(\tilde{\mathbf{z}}', p')$. Finally, the model owner will attempt to detect whether the forged image is watermarked by applying the inverse diffusion process with the private model weights to the forged image. Let $\tilde{\mathbf{z}}^{\text{attack}} = G_\Theta^{-1}(\texttt{image'})$. As an upper bound on the capability of this attack, we perform it with the same prompt. Also, Keles & Hegde (2023) demonstrates that inverting a generative model is a significantly challenging task.

Strikingly, we find that the similarity between the true noise and the noise reconstructed with the model weights is almost always greater than a relatively large threshold $\tau = 0.5$ ($p$ value $< 10^{-3}$, Figure 2). At the same time, the reconstructed similarity from the image made by an attacker using the reconstruction attack $\texttt{sim}(\mathbf{z}_{i^*}, \tilde{\mathbf{z}}^{\text{attack}})$, along with the similarity to random vectors $\texttt{sim}(\mathbf{z}_{i^*}, \mathbf{z})$ are both much smaller. Namely, they are respectively $z = 5.3$ and $z = 9.4$ standard deviations

away from the mean (Table 4). Taken together, these results mean that the probability $p$ of a non-watermarked image mistakenly labeled as watermarked is very low in both cases. For the random noise, the probability to confuse it is as the initial noise is $p < e^{\left(\frac{\tau^2}{2\sigma^2}\right)} < 10^{-19}$, allowing practically a perfect distinction between any pair of unrelated noises.

**Runtime considerations.** Our method requires searching over all $N$ watermarks, leading to a naive runtime complexity of $\mathcal{O}(N)$. However, more efficient algorithms for similarity-based search, such as HNSW (Malkov & Yashunin, 2018), can reduce this complexity to $\mathcal{O}(\log N)$, at the expense of additional memory usage. We provide empirical runtime analysis of our method in Appendix H. For large enough values of $N$, this cost may eventually become undesirable. Together with our aim to maintain high robustness with an increasing number of keys, it motivates a more efficient method, which is presented in the next section.

## 4 Method

### 4.1 WIND: Two-stage Efficient Watermarking

While always using a single initial noise for our model might imply good robustness properties, to make forgery and removal more difficult, it is generally preferable to maintain a large set of $N$ initial noises to be used by the model. More importantly, using a large number of different noises $N$ may serve as different keys, encoding some metadata about each image. This metadata might include information about the specific model that generated it, as well as additional information about the generation for further validation of the image source, once detected.

In order to make the search over a large number of noises more efficient, we introduce a two-stage efficient watermarking approach we name **WIND** (**W**atermarking with **I**ndistinguishable and Robust **N**oise for **D**iffusion Models). First, we initialize $M$ groups of initial noise, each group associated with its own Fourier-pattern key. In contrast to prior work, we employ these Fourier patterns not as a watermark, but as a *group identifier* to reduce the search space.

For each image generation, we randomly select an index for the initial noise, denoted as $i^* \in [N]$. We use a group identifier $g^* = i^* \% M$, where $\%$ denotes the modulus operation. We embed $g^*$ in the Fourier space of the latent noise (similar to Wen et al. (2023)). During detection, we reconstruct the latent noise and find the group identifier $\tilde{g}$ that is closest to the Fourier pattern embedded in the image. We then search over all indices $i$ such that $\tilde{g} = i \% M$. In this way, the search space has a size of $N/M$ rather than $N$. We include an algorithm box for generation (Algorithm 1) and detection (Algorithm 2).

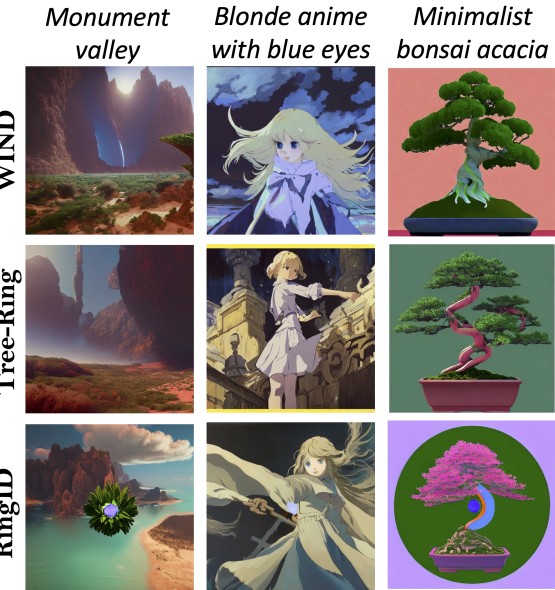

Figure 4: Qualitative results of watermarked images generated using WIND, Tree-Ring, and RingID. See Appendix D for quantitative results. See Appendix J for additional qualitative results.

Table 1: Comparison of correct watermark detection accuracy between WIND and previous image watermarking approaches under various image transformation attacks. $\text{WIND}_M$ denotes the use of $M$ groups, with the total number of noises ($N$) specified in the "Keys" column. A broader comparison with additional methods can be found in Table 16.

| Method | Keys | Clean | Rotate | JPEG | C&S | Blur | Noise | Bright | Avg ↑ |
|---|---|---|---|---|---|---|---|---|---|
| Tree-Ring | 32 | 0.790 | 0.020 | 0.420 | 0.040 | 0.610 | 0.530 | 0.420 | 0.404 |
| | 128 | 0.450 | 0.010 | 0.120 | 0.020 | 0.280 | 0.230 | 0.170 | 0.183 |
| | 2048 | 0.200 | 0.000 | 0.040 | 0.000 | 0.090 | 0.070 | 0.060 | 0.066 |
| RingID | 32 | **1.000** | **1.000** | **1.000** | 0.530 | 0.990 | **1.000** | 0.960 | 0.926 |
| | 128 | **1.000** | 0.980 | **1.000** | 0.280 | 0.980 | **1.000** | 0.940 | 0.883 |
| | 2048 | **1.000** | 0.860 | **1.000** | 0.080 | 0.970 | 0.950 | 0.870 | 0.819 |
| $\text{WIND}_{\text{fast}_{128}}$ | 100000 | **1.000** | 0.780 | **1.000** | 0.470 | **1.000** | **1.000** | 0.960 | 0.887 |
| $\text{WIND}_{\text{fast}_{2048}}$ | 100000 | **1.000** | 0.870 | 0.960 | 0.060 | 0.960 | 0.950 | 0.900 | 0.814 |
| $\text{WIND}_{\text{full}_{128}}$ | 100000 | **1.000** | 0.780 | **1.000** | 0.850 | **1.000** | **1.000** | **1.000** | 0.947 |
| $\text{WIND}_{\text{full}_{2048}}$ | 100000 | **1.000** | 0.880 | **1.000** | **0.930** | **1.000** | 0.990 | 0.980 | **0.969** |

In the following part, we refer to two variants of our method: (i) **_WIND_**_fast_ where we assume the used initial noise belongs to the identified group $\tilde{g}$ and check similarity only to noise patterns in this group. (ii) **_WIND_**_full_ where we check all $N$ possible initial noises if we can't find a match within the detected group (the gap between the similarity of the correct noise and random noises, as shown in Figure 2, allows us to determine whether the correct noise has been identified). This method is slower but more robust to removal attacks that might interfere with the Fourier pattern. Additional ablations and results can be found in Appendix D.

## 4.2 RESILIENCE TO FORGERY

In addition to empirical evaluations of specific attacks as in Figures 2 and 3; we discuss below the attacker's ability to infer knowledge about the used noise pattern across different watermarked images. Even if the attacker is able to obtain information about a specific initial noise $\mathbf{z}_i$ for an index $i$ (which is an extreme case), the other noise vectors for $j \neq i$ are still safe[1]. This is because we use a cryptographic hash function and a secret salt. Formally, Theorem 4.1 shows that, as long as the cryptographic hash function remains unbroken and the secret salt is kept private, the watermarking algorithm maintains its security properties against even very powerful adversaries.

**Theorem 4.1.** _[Cryptographic Security] Let_ `hash`: $0, 1^* \to 0, 1^\ell$ _be an unbroken cryptographic hash function used in our watermarking algorithm, with inputs_ $i^* \in [N]$ _and a secret salt_ $s$. _Assume_ $s$ _is sufficiently long and randomly generated. Then, even if an adversary obtains: the group number_ $g^*$, _the initial noise index_ $i^*$, _the initial noise_ $\mathbf{z}_{i^*}$, _and even the corresponding output of the hash function_ `seed`, _the adversary cannot:_

1. _Recover the secret salt_ $s$,

2. _Generate valid reconstructed noise_ $\mathbf{z}_j$ _for any other initial noise index_ $j \neq i$

We defer the proof to Appendix E.

---

[1]We note that obtaining a single noise pattern might not be enough to effectively forge the watermark, as the model owner may encode this pattern with additional metadata as described in Section 4.1

Table 2: Cosine similarity between the initial noise and the inversed noise before and after the regeneration attack. Also see Appendix D

| Condition | Mean | STD |
|---|---|---|
| Original Image | 0.888 | 0.053 |
| Attacked Image | 0.824 | 0.062 |
| Unrelated Image | 0.000 | 0.008 |

Table 3: FID scores of WIND compared to previous watermarking approaches.

| Method | FID ↓ |
|---|---|
| DwtDctSvd | 25.01 |
| RivaGAN | 24.51 |
| Tree-Ring | 25.93 |
| RingID | 26.13 |
| WIND | **24.33** |

### 4.3 Watermarking Non-Synthetic Images.

Until now, we have addressed watermarking only for AI-generated synthetic images. Yet, protecting copyrights, or preventing the spread of misinformation, may also apply to modified natural images. Most previous approaches to watermark diffusion models overlook attempting to expand their method to non-generated images. To allow using our framework for non-generated images, we expand our framework. By using diffusion inpainting, our watermark can be applied to a natural image. Later, by inverting the inpainted image we can verify the presence of the watermark.

As demonstrated in Figure 5, our inpainting method injects a watermark with minimal visual impact, preserving the original image's integrity. Please see Appendix D for additional results.

## 5 Experiments

### 5.1 Watermark Robustness

**Setting.** For a fair comparison with previous methods (Ci et al., 2024; Wen et al., 2023), we employed Stable Diffusion-v2 (Rombach et al., 2022), with 50 inference steps for both generation and inversion. Other implementation-details can be found in **??**.

**Image Transformation Attacks.** Following previous methods (Wen et al., 2023; Ci et al., 2024) we applied these image transformations to the generated images: 75° rotation, 25% JPEG compression, 75% random cropping and scaling (C & S), Gaussian blur with an $8 \times 8$ filter size, Gaussian noise with $\sigma = 0.1$, and color jitter with a brightness factor uniformly sampled between 0 and 6. In Table 1 we compare our methods to both Tree-Ring and RingID. As the results demonstrate, using multiple keys with RingID (Ci et al., 2024) is possible. Yet, it remains vulnerable to cropping and scaling attacks. In contrast, WIND effectively addresses this challenge. It enables accurate watermark detection under all image transformation attacks. We note that the incorporation of the keys in the RingID method not only allows us to embed keys but also increases the robustness of the full method to certain attacks.

**Steganalysis Attack.** We assess the robustness of our method against the attack proposed by Yang et al. (2024a), which is capable of forging and removing the Tree-Ring and RingID keys. As discussed in Section 2.3, this attack attempts to approximate the watermark by subtracting watermarked images from non-watermarked images. The results, presented in Figure 3, indicate that while the attack could be able to forge or remove our group identifier, it is unable to forge or remove our watermark (initial noises). Even when the Fourier pattern type key is removed through an exhaustive search, our method remains robust in identifying the correct initial noise.

Table 4: Cosine similarity between the first initial noise used for generation and the inversed noise obtained through three inversion approaches. "Private" refers to models owner's model, while "Public" denotes external model.

| Approach | Mean | Std |
|---|---|---|
| Gen (private) → Rev (private) | 0.888 | 0.053 |
| Gen (private) → Rev (public) → Gen (public) → Rev (private) | 0.166 | 0.063 |
| Random Noise | 0.000 | 0.053 |

**Regeneration Attacks.** Recently, Zhao et al. (2023a) introduced a two-stage regeneration attack: (i) adding noise to the representation of a watermarked image, and (ii) reconstructing the image from this noisy representation. To assess the resilience of our approach to regeneration attacks, we applied the attack from Zhao et al. (2023a) to watermarked images generated by our model. As shown in Table 2, the attack has a minimal impact on the distribution of the cosine similarities between the initial noise and the inverted noise. The attacked noise similarity still maintains a significant gap compared to random noise.

To examine the performance of our inpainting method, we report the Fréchet Inception Distance (FID) (Heusel et al., 2018) on the MS-COCO-2017 (Lin et al., 2015) training dataset in Table 3. Notably, our method achieves the lowest FID among the compared methods, indicating a closer alignment with real images. Additionally we include some images generated by our framework in Figure 4.

## 6 Discussion and Limitations

**Editing a Given Image vs. Forging.** While forging our watermark by obtaining the initial noise is hard (Section 3), an easier path to obtaining harmful watermarked images might be to apply a slight edit to an already watermarked image. An harmful image in this context might include a copy-right infringing image, NSFW image, or any other content the model owner wish to avoid being associated with. Naturally, there is a trade-off between the severity of the applied edit, and the edit ability to preserve the initial watermark. We present one solution to mitigating this issue in the next discussion point.

**Storing a Database of Generations.** Model owners wishing to protect themselves from an attacker modifying a watermark image may keep a database of the past generations by their model. For these extreme cases, the model owner might only save the used prompts and initial noiseseeds, and use the reconstructed noise to retrieve the entire set of prompts used with that specific seed (Huang & Wan, 2024). While this process may be resource-intensive, it is only required in the rare event that an attacker intentionally modifies a benign image into a harmful one while preserving the watermark.

**Private Model.** Our watermark robustness is based to a large extent on the inability of an attacker to invert a model, which is empirically validated but not mathematically proven. Yet, as discussed in Section 2.2, the ability to successfully invert our model may be nearly equivalent to the ability to steal the forward diffusion process, effectively stealing the model (in which case, any watermarking attempt might be deemed quite useless anyhow). Still, a better framing of the mathematical assumptions behind this claim is a limitation of this work, as well as of previous works on watermarking using inversion of the diffusion generative process.

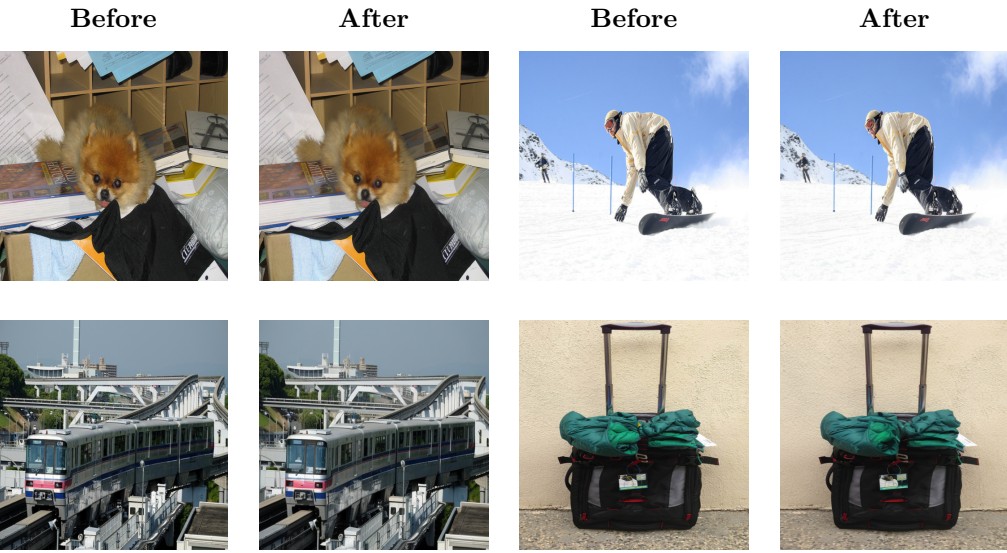

| Before | After | Before | After |

Figure 5: Comparison of COCO images before and after watermarking via inpainting.

**Attacker's Advantage.** There exists a large set of diverse attacks aimed at watermark removal (Zhang et al., 2023; Yang et al., 2024a; Zhao et al., 2023a), along with image transformations such as rotation and crops that also achieve some limited success against our watermark. As in many security applications, we suspect that an attacker capable enough will still be able to remove the watermark using new techniques we might not expect. However, a more robust watermark may nevertheless help to decrease the spread of false information.

Additional discussion and limitations can be found in Appendix C.

## 7 CONCLUSION

In this work, we present a robust and distortion-free watermarking method that leverages the initial noises employed in diffusion models for image generation. By integrating existing techniques, we enhanced the approach to achieve improved efficiency and robustness against various types of attacks. Furthermore, we outlined a strategy for applying our method to non-generated images through inpainting.

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

# Appendices

# Contents

# A NOTATION

Table 5: Notations used in the paper.

| | |
|---|---|
| $N$ | Number of initial noises |
| $M$ | Number groups |
| $s$ | Secret salt for cryptographic security |
| $i$ | Index of initial noise: $i \in [N]$ |
| $g$ | Index of group: $g = i\%M$ |
| hash | A cryptographic hash function |
| $\mathbf{z}$ | Initial noise |
| $\tau$ | Threshold for declaring an image is watermarked |
| $T$ | Number of diffusion steps |
| $\Theta$ | Weights of a diffusion model |
| $p$ | Text prompt for diffusion |
| $G_\Theta$ | Diffusion model with weights $\Theta$ |
| $G_\Theta^{-1}$ | Inverse diffusion model with weights $\Theta$ |

# B RELATED WORKS

**Memorization in Diffusion Models.** Diffusion models (Ho et al., 2020; Sohl-Dickstein et al., 2015) have demonstrated a capacity not only to generalize but also to memorize training data. This can lead to the reproduction of specific patterns or, in some cases, exact content from the training set, including sensitive or proprietary information. This memorization poses significant risks of unintended intellectual property leakage, particularly in large-scale generative models. Several studies have shown that information from training data can be extracted from diffusion models (Carlini et al., 2023b; Somepalli et al., 2023b; Carlini et al., 2023a; Gu et al., 2023; Somepalli et al., 2023a).

**Image Watermarking.** Image watermarking is essential for protecting intellectual property, verifying content authenticity, and maintaining the integrity of digital media. The field has ranged from traditional signal processing techniques to recent deep learning methods (Potdar et al., 2005; Singh & Singh, 2023).

Among Early watermarking strategies, one of the simplest methods was Least Significant Bit (LSB) embedding, which modifies the least significant bits of image pixels to imperceptibly embed watermarks (Wolfgang & Delp, 1996). Another classical approach utilized frequency-domain transformations and Singular Value Decomposition (SVD) to hide watermarks within image coefficients. (Chang et al., 2005; Al-Haj, 2007).

Recent developments leverage deep learning for watermarking. For instance, HiDDeN (Zhu et al., 2018) introduced an end-to-end trainable framework for data hiding. RivaGAN (Zhang et al., 2019) utilizes adversarial training to embed watermarks, while Lukas & Kerschbaum (2023) proposed an embedding technique that optimizes efficiency by avoiding full generator retraining.

Table 6: Inpainting correct watermark detection accuracy.

| Clean | Rotate | JPEG | C&S | Blur | Noise | Bright | Avg ↑ |
|-------|--------|------|-----|------|-------|--------|-------|
| 1.000 | 1.000 | 1.000 | 0.880 | 1.000 | 0.950 | 0.950 | 0.969 |

**Watermarking for Diffusion Models.** Existing watermark methods for diffusion models can be divided into three categories:

(i) Post-processing methods which adjust image features to embed watermarks (Zhao et al., 2023c; Fernandez et al., 2023b). This approach alters the image and its distribution, which can result in significant changes to the generated image. However, recent work by Zhao et al. (2023b) shows that pixel-level perturbations are removable by regeneration attacks makes. To date, this approach is not robust.

(ii) Fine-tuning-based approaches combine the watermark within the generation process (Zhao et al., 2023c; Xiong et al., 2023; Liu et al., 2023; Fernandez et al., 2023a; Cui et al., 2023). To date, these methods have robustness issues as well (Zhao et al., 2023a).

(iii) Tree-Ring introduced an approach to proposing a method to imprint a tree-ring pattern into the initial noise of a diffusion model (Wen et al., 2023). Each pattern is used as a key, which is added in the Fourier space of the noise. The verification of the presence of the key involves recovering the initial noise from the generated image and checking if the key is still detectable in Fourier space. This approach makes Tree-Ring and its follow-up works the most robust approach against attacks (Zhao et al., 2023a; An et al., 2024).

Recently, Yang et al. (2024a) took advantage of the distribution shift present in Tree-Ring that occurs with impainting keys and arranged the first successful black box attack against it, as we detailed in Section 2.3.

## C   ADDITIONAL DISCUSSION AND LIMITATIONS

**Relation to Other Initial Noise Watermarking Methods.** The seminal work by Wen et al. (2023) innovated the use of initial noise in DDIM for watermarking. Most related to our work, Yang et al. (2024b) also embeds a watermark in the initial noise already used by a DDIM diffusion model. Yet, while Yang et al. (2024b) proposes a watermark that is distortion-free for a single image, it is not distortion-free when examining sets of images; therefore it is vulnerable to attacks such as Yang et al. (2024a). We aim to be robust to attacks even when many images are examined together.

There are additional technical differences between our approach and Yang et al. (2024b). Most notably: (i) Our work also studies applying our watermark to non-synthetic (natural) images, or images coming from other generative models. (ii) While Yang et al. (2024b) design a function to embed specific bits into the initial noise, we take another approach. Namely, we view the entire initial noise (with generation and inversion) as a noisy channel. Inspired by Shannon (1949), we use a random encoding of the watermark identities into the channel.

**Computational Requirements.** As discussed in Section 3 our similarity search can be accelerated given well-known methods. Yet, the computational requirements of our method might be limiting when trying to use our method on edge devices. However, similarly to Tree-Ring and Ring-ID (Wen et al., 2023; Ci et al., 2024) our method assumes a private model, which is usually not deployed on edge devices anyhow.

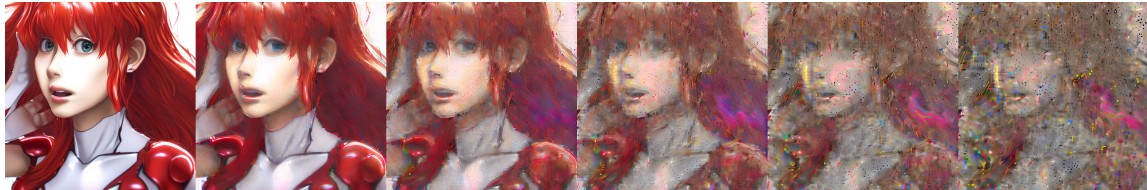

Figure 6: Image sequence from 0 to 50 regeneration attack iterations.

**Trade-Offs Between the Watermarking Overhead, and Detection Accuracy.** We suggest the following variants of our method for different possible requirements of runtime scaling, detection robustness, and ease of adaptation.

A. Detection of the group identifier alone: This operation takes a search of $\mathcal{O}(M)$, but is vulnerable to both removal and forgery attempts, as we use a weaker watermark for group identifiers.

B. Detection of the Fourier pattern, followed by a validation of the exact initial noise (WIND$_{\text{fast}}$): within the group. This operation takes $\mathcal{O}(N/M)$ search. It is vulnerable to removal attempts, but more resilient to forgery attempts (see Table 1).

C. An exhaustive search of the initial noise, also outside the identified group (WIND$_{\text{full}}$): This operation takes $\mathcal{O}(N)$ search. It is more resilient to both removal and forgery attempts (see Figure 3, and Table 1).

This method, while slower is also easier to adapt. A user that wishes to use a fast version of this variant may apply a similar algorithm to the one described above using only a few possible random noises. This would replace the distinguishability of many different watermarks with the ability to rapidly and simply detect the watermarked images.

Practically, an NN search can be accelerated using many methods and can be scaled to tens of millions without significantly affecting the detection time (Wei et al., 2024; Douze et al., 2024; Malkov & Yashunin, 2018; Andoni et al., 2018).

**Image Quality Considerations.** Our method relies on using an initial random noise, drawn from the same distribution of initial noises already used by the model. Therefore, the core of our method (the initial noise stage) is not compromising the visual quality of the generated images at all.

The only effect on visual quality comes from the group identifier, where we use existing off-the-shelf watermarking images. In our implementation, we used the RingID (Ci et al., 2024) method that adds the Fourier pattern to the initial noise.

When a model owner wishes to preserve image quality even better, they may use any other existing watermarking method for the group identifier stage. This will still not compromise the security provided by the initial noise stage.

**Inversion Attack.** As discussed in Section 2.2 in our paper, accurately inverting the model is as difficult as copying the forward process of the model (image generation). While hard, an attacker able to do so is effectively also capable of generating novel images using the same diffusion process. Therefore, At this stage, the model itself is effectively compromised (and not only the watermark signature). We believe that being as hard to forge as the model itself, is a reasonable level of security for almost all use cases.

Table 7: Impact of different inference steps on detection accuracy.

| Steps | Clean | Rotate | JPEG | C&S | Blur | Noise | Bright | Avg ↑ |
|-------|-------|--------|------|-----|------|-------|--------|-------|
| 20 | **1.000** | 0.780 | **1.000** | 0.880 | 0.920 | **1.000** | 0.960 | 0.934 |
| 50 | **1.000** | **0.930** | **1.000** | **0.940** | **1.000** | 0.980 | 0.980 | 0.976 |
| 100 | **1.000** | **0.930** | **1.000** | **0.940** | **1.000** | **1.000** | 0.990 | **0.980** |
| 200 | **1.000** | 0.850 | **1.000** | **0.940** | **1.000** | **1.000** | **1.000** | 0.970 |

Yet, approximately inverting the model might also be a threat. While even approximately inverting a model is also very hard, it might be easier than stealing the model. Still, we would like to emphasize that our method is more secure than other diffusion-process-based watermarking techniques, where image distortion themselves may allow easier forging (Yang et al., 2024a).

## D   ADDITIONAL RESULTS

### D.1   APPLICABILITY TO OTHER TYPES OF MODELS

We expect our watermark to be effective directly for any model for which some inversion to the original noise is possible. Namely, as the correlation between random noises in a very high dimension is very much concentrated around 0, even a very slight success in the inversion process is enough to be distinguishable. In higher generation resolutions the dimensionality of the noise is even higher, and therefore the separation would be even better (El Karoui, 2009).

Empirically, to validate the generality of our method, we also report results for the SD 1.4 model (Rombach et al., 2022). Using $N = 10000$ noises and $M = 2048$ group identifiers, our method achieved a detection accuracy of **97%** to identify the correct watermark (initial noise).

In any case, our method of the reported SD 2.1 model can also be used to watermark images collected from other sources (please see Section 4.3, Appendix D.2).

### D.2   NON-SYNTHETIC IMAGES WATERMARK DETECTION

Our inpainting method allows us to watermark both images generated by any model and non-generated images. To evaluate the robustness of the inpainting watermarking approach, we present results in Table 1 for this method, utilizing $N = 100$ noises. Results are shown in Table 6.

### D.3   FURTHER EXPLORATION OF THE REGENERATION ATTACK PERTURBATION STRENGTH

In Section 5.1, we discussed the robustness of WIND against regeneration attacks. However, using it iteratively might still be a stronger adversary. We applied the regeneration attack proposed by Zhao et al. (2023a), up to 50 times. We see that iterative regeneration indeed decreases the similarity between the original noise and the reconstructed one. This happens as the image becomes less and less correlated to the original generation Figure 6.

Yet, the detection rate of our algorithm remains very high Table 14. We attribute this to the fact that even a slight remaining correlation between the attacked image and the initial noise remains significant with respect to the correlation expected from non-watermarked images. This happens because of the very low correlation between random (non-watermarked) noises (Figure 2).

### D.4 Quantitative Analysis of the Effect on Image Quality

We reported the FID of our model on Table 3. To further assess the effect of WIND watermark on image quality we report the CLIP score Hessel et al. (2021) before and after watermarking on Table 10. Results indicate that adding the watermark has a negligible effect on the CLIP score for generated images.

To further quantify the distortion introduced by each model, we report pixel-base matrices, SSIM and PSNR in the two settings we study:

**Images Generated by the Diffusion Model.** WIND's distortion arises from using *group identifiers*, enabling faster detection. To disentangle this effect, we also evaluate $WIND_{w/o}$, which omits group identifiers. As can be seen in Table 11, the image quality generated using our full method is comparable to that of previous techniques. Users who wish to generate distortion-free images, without affecting image quality, can do so by omitting the group identifier (at the cost of a slower detection phase for very large values of $N$).

**Watermarking Non-Synthetic Images.** Additionally, we present results for $WIND_{inpainting}$, our inpainting-based approach capable of watermarking both non-synthetic images and outputs from other generative models (Table 15). Although other watermarking methods may preserve image quality better, our image quality remains high. Importantly, to the best of our knowledge, our approach is the only one capable of watermarking non-synthetic images while remaining robust against the regeneration attack (Zhao et al., 2023a). Therefore, it is preferable when an adversary may try to remove the watermark.

In addition, the inpainting technique can be applied selectively to specific parts of the image if the copyright owner wishes to perfectly preserve fine details in certain areas.

### D.5 Robustness Comparison to Different Number of Inference Steps

We evaluate the impact of inference steps on detection accuracy, as shown in Table 7. The results indicate that using 100 steps yields better detection accuracy compared to other step counts, including the 50 steps used in our main experiments.

Table 8: Error bars of WIND.

| AUC | TP@1% |
|-------|-------|
| 0.971 | 1.000 |

### D.6 True Positive and AUC

Expanding on the detection assessment settings discussed in Section 5, we reported WIND's error bars. AUC and True Positive (TPR@1%FPR) results are available on Table 8. Demonstrate strong performance, emphasizing WIND's robustness and reliability.

### D.7 Evaluation Against Additional Attacks

We evaluate WIND against a diverse set of attacks, including transfer-based, query-based, and white-box methods. Specifically, we employ the WeVade white-box attack (Jiang et al., 2023), the transfer attack described in Hu et al. (2024), a black-box attack utilizing NES queries (Ilyas et al., 2018), and a random search approach discussed in Andriushchenko et al. (2024), adopted to

Table 9: Success rate of additional attacks.

| WeVade | Random Search | Transfer Attack | NES Query |
|--------|---------------|-----------------|-----------|
| 1% | 2% | 3% | 2% |

attempt watermark removal. The success rates of these attacks are detailed in Table 9. Notably, none of these methods succeed against WIND, as the correct watermark remains detectable in over 97% of cases even after applying these attacks.

## E    Proof of Resilience to Forgery

The WIND method is an approach for generating multiple watermarked images. Theorem 4.1 tells us that compromising one or more watermarked images does not give away any information about any other watermarked images. E.g., the adversary cannot "generate valid reconstructed noise for any other initial noise index $j \neq i$". That said, Theorem 4.1 does leave open the possibility that an adversary can take a watermark image, reconstruct the initial noise only for that image, and use it to attack the method, which we evaluate empirically.

**Cryptographic Background**   Consider a cryptographic hash function $\texttt{hash}: \{0,1\}^* \to \{0,1\}^\ell$ with $\ell$ output bits. E.g., $\ell = 256$ for SHA-256. We will describe properties of the hash function in terms of 'difficulty'; we say a task is 'difficult' if, as far as we know, finding a solution is almost certainly beyond the computational capabilities of any reasonable adversary. An *unbroken* cryptographic hash function satisfies the following properties: *Pre-image resistance* requires that given a hashed value $v$, it is difficult to find any message $m$ such that $v = \texttt{hash}(m)$. *Second pre-image resistance* requires that given an input $m_1$, it is difficult to find a different input $m_2$ such that $\texttt{hash}(m_1) = \texttt{hash}(m_2)$. *Collision resistance* requires that it is difficult to find two different messages $m_1$ and $m_2$ such that $\texttt{hash}(m_1) = \texttt{hash}(m_2)$.

**Theorem 4.1.** *[Cryptographic Security] Let* $\texttt{hash}: 0,1^* \to 0,1^\ell$ *be an unbroken cryptographic hash function used in our watermarking algorithm, with inputs $i^* \in [N]$ and a secret salt $s$. Assume $s$ is sufficiently long and randomly generated. Then, even if an adversary obtains: the group number $g^*$, the initial noise index $i^*$, the initial noise $\mathbf{z}_{i^*}$, and even the corresponding output of the hash function* $\texttt{seed}$, *the adversary cannot:*

   1. *Recover the secret salt $s$,*

   2. *Generate valid reconstructed noise $\mathbf{z}_j$ for any other initial noise index $j \neq i$*

*Proof of Theorem 4.1.* We will prove each part of the theorem separately:

1. The adversary cannot recover the secret salt $s$: Given the output seed $= \texttt{hash}(i^*, s)$ and partial input $i^*$ the adversary aims to find $s$. This is equivalent to finding a pre-image of given partial information about the input. By the pre-image resistance property of cryptographic hash functions, this task is computationally infeasible. Even if the adversary knows all possible values of $i$, the space of possible secret salts $s$ is too large to search exhaustively (as $s$ is a sufficiently long random string). Therefore, the adversary cannot recover $s$.

2. The adversary cannot generate valid reconstructed noise for any other initial noise index $j \neq i$. This security guarantee is ensured by two properties of $\texttt{hash}$: a) Second pre-image resistance: Given

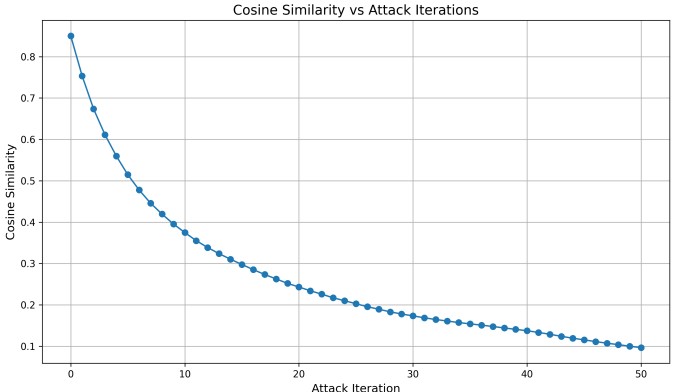

Figure 7: Cosine Similarity from 0 to 50 regeneration attack iterations.

$(i^*, s)$, it's computationally infeasible to find $(i', s)$ where $i' \neq i^*$ such that $\mathtt{hash}(i^*, s) = \mathtt{hash}(i', s)$.
b) Collision resistance: It's computationally infeasible to find any two distinct inputs that hash to the same output. These properties ensure that the adversary cannot find alternative inputs that produce the same hash output, and thus cannot generate valid reconstructed noise for different index numbers $j$. □

Table 10: Effect of WIND on CLIP score.

| CLIP Before Watermark | CLIP After Watermark |
|:---:|:---:|
| 0.366 | 0.360 |

Theorem 4.1 leaves open the possibility that an adversary can recover the noise from a watermarked image and use *that* noise to forge a new watermarked image. However, empirically we show that this attack fails without access to the weights of the private diffusion model.

Table 11: SSIM and PSNR values of initial noise-based watermarking approaches. $\mathrm{WIND}_{\mathrm{w/o}}$ refers to the method without *group identifiers*

| Method | SSIM ↑ | PSNR ↑ |
|:---|:---:|:---:|
| $\mathrm{WIND}_{\mathrm{w/o}}$ | **1.000** | $\infty$ |
| $\mathrm{WIND}_{\mathrm{full}}$ | 0.494 | 14.647 |
| RingID | 0.454 | 13.560 |
| Tree-Ring | 0.545 | 15.251 |

## F  FURTHER DISCUSSION ON DISTORTION

Using the same initial noise for multiple generations is not distortion-free when examining groups of images. For example, all images with the same prompt $p$ and the same initial noise $z$ will

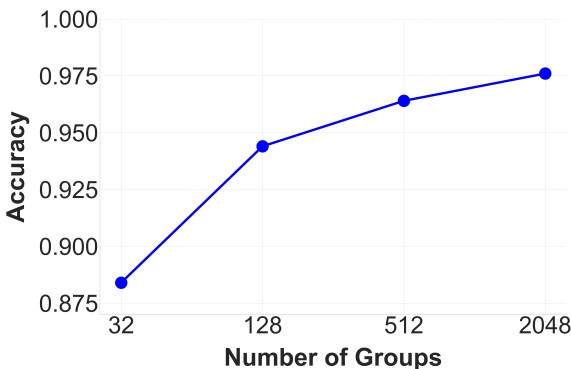

Figure 8: Effect of the number of groups on the average accuracy of retrieving initial noise from 10,000 samples under various image transformation attacks.

be identical, distorted away from the distribution of groups of images generated with i.i.d noises. Luckily, the huge gap between the similarities distribution of (i) reconstructed vs. used noise and (ii) reconstructed vs. another noise, allows us to use as many different noise patterns, while still keeping the noise we used distinguishable more similar to the reconstructed noises. Therefore, limiting the level of distortion in practice.

Table 12: Detection time (second)

| WIND | Tree-Ring | RingID |
|------|-----------|--------|
| 22 | 20 | 14 |

Table 13: Accuracy of retrieving the initial noise from 10,000 noise samples, divided into varying numbers of groups, under different image transformation attacks.

| Groups | Clean | Rotate | JPEG | C&S | Blur | Noise | Bright | Avg ↑ |
|--------|-------|--------|------|-----|------|-------|--------|-------|
| 32 | **1.000** | 0.540 | **1.000** | 0.700 | **1.000** | 0.990 | 0.960 | 0.884 |
| 128 | **1.000** | 0.810 | **1.000** | 0.820 | **1.000** | **1.000** | 0.980 | 0.944 |
| 512 | **1.000** | 0.890 | **1.000** | 0.880 | **1.000** | 0.980 | **1.000** | 0.964 |
| 2048 | **1.000** | **0.930** | **1.000** | **0.940** | **1.000** | 0.980 | 0.980 | **0.976** |

## G  NUMBER OF GROUPS

In our framework, we divide the initial noises into $N$ groups and associate a Tree-Ring-type key with each group. The use of Fourier Pattern keys enables robustness against rotation, and grouping reduces the search space for inverted noise.

To investigate the impact of the number of groups, we performed an experiment with 10,000 noises and varied the number of groups from 32 to 2048. As expected, Figure 8 demonstrates that increasing the number of groups leads to better accuracy in detecting the correct initial noise. This is because a larger number of groups results in fewer noises per group, which facilitates more

Table 14: Correct watermark detection after iterative regeneration attack.

| Iteration | Cosine Similarity | Detection Rate |
|-----------|-------------------|----------------|
| 10 | 0.493 | 100% |
| 20 | 0.342 | 100% |
| 30 | 0.243 | 100% |
| 40 | 0.170 | 100% |
| 50 | 0.121 | 100% |

Table 15: SSIM and PSNR values for non-synthetic image watermarking approaches.

| Method | SSIM ↑ | PSNR ↑ |
|--------|--------|--------|
| WIND$_{\text{inpainting}}$ | 0.768 | 26.806 |
| DwtDctSvd | 0.983 | 39.381 |
| RivaGAN | 0.978 | 40.550 |
| SSL | **0.984** | 41.795 |
| StegaStamp | 0.911 | 28.503 |

accurate detection. Detailed results for each number of groups under transformation attacks are reported in Table 13.

## H  Empirical Runtime Analysis

However, the runtime is highly sensitive to the available computational resources. To provide a practical estimate, we measured the detection time using a single NVIDIA GeForce RTX 3090. Specifically, we divided 100,000 initial noise samples into 32 groups and reported the detection. Under these conditions, the detection phase for 100,000 noise samples takes approximately 22 seconds per detection. We include a comparison with other methods in Table 12.

## I  Implementation Details

**Prompts.**  For all evaluations we used the set of prompts taken from Gustavosta (2024).

**Threshold for Detection.**  For the first variant $WIND_{fast}$ (see Section 4) we use a threshold of min $\ell_2$ norm $> 160$. The second variant ($WIND_{full}$) does not use a threshold, but rather, we choose the noise pattern within the group that has the lowest $\ell_2$ as our candidates for the identified noise.

**General Retrieval Details.**  We included simple rotation (using intervals of 2 degrees) and sliding window (window size of 32, stride of 8) searches as part of the retrieval process. These searches do not involve directly optimizing for the specific degrees of rotation or cropping encountered, ensuring that robustness remains intrinsic to the method.

Table 16: Comparison of correct watermark detection accuracy between WIND and previous image watermarking approaches under various image transformation attacks. $\text{WIND}_M$ denotes the use of $M$ groups, with the total number of noises $(N)$ specified in the "Keys" column.

| Method | Keys | Clean | Rotate | JPEG | C&S | Blur | Noise | Bright | Avg ↑ |
|---|---|---|---|---|---|---|---|---|---|
| DwtDct | 1 | 0.974 | 0.596 | 0.492 | 0.640 | 0.503 | 0.293 | 0.519 | 0.574 |
| DwtDctSvd | 1 | **1.000** | 0.431 | 0.753 | 0.511 | 0.979 | 0.706 | 0.517 | 0.702 |
| RivaGan | 1 | 0.999 | 0.173 | 0.981 | **0.999** | 0.974 | 0.888 | 0.963 | 0.854 |
| | 32 | 0.790 | 0.020 | 0.420 | 0.040 | 0.610 | 0.530 | 0.420 | 0.404 |
| Tree-Ring | 128 | 0.450 | 0.010 | 0.120 | 0.020 | 0.280 | 0.230 | 0.170 | 0.183 |
| | 2048 | 0.200 | 0.000 | 0.040 | 0.000 | 0.090 | 0.070 | 0.060 | 0.066 |
| | 32 | **1.000** | **1.000** | **1.000** | 0.530 | 0.990 | **1.000** | 0.960 | 0.926 |
| RingID | 128 | **1.000** | 0.980 | **1.000** | 0.280 | 0.980 | **1.000** | 0.940 | 0.883 |
| | 2048 | **1.000** | 0.860 | **1.000** | 0.080 | 0.970 | 0.950 | 0.870 | 0.819 |
| $\text{WIND}_{\text{fast}_{128}}$ | 100000 | **1.000** | 0.780 | **1.000** | 0.470 | **1.000** | **1.000** | 0.960 | 0.887 |
| $\text{WIND}_{\text{fast}_{2048}}$ | 100000 | **1.000** | 0.870 | 0.960 | 0.060 | 0.960 | 0.950 | 0.900 | 0.814 |
| $\text{WIND}_{\text{full}_{128}}$ | 100000 | **1.000** | 0.780 | **1.000** | 0.850 | **1.000** | **1.000** | **1.000** | 0.947 |
| $\text{WIND}_{\text{full}_{2048}}$ | 100000 | **1.000** | 0.880 | **1.000** | 0.930 | **1.000** | 0.990 | 0.980 | **0.969** |

# J    ADDITIONAL QUALITATIVE RESULTS

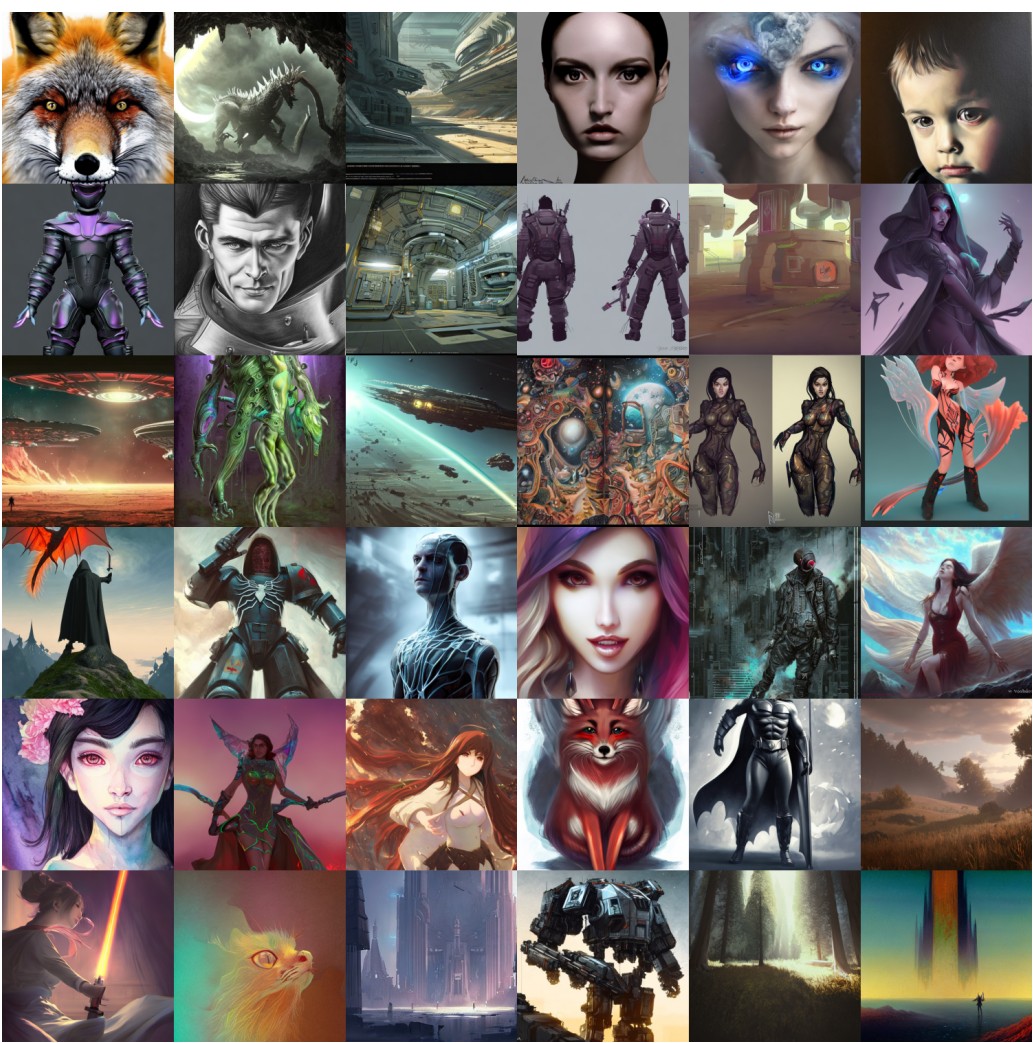

Figure 9: More watermarked images generated with WIND.

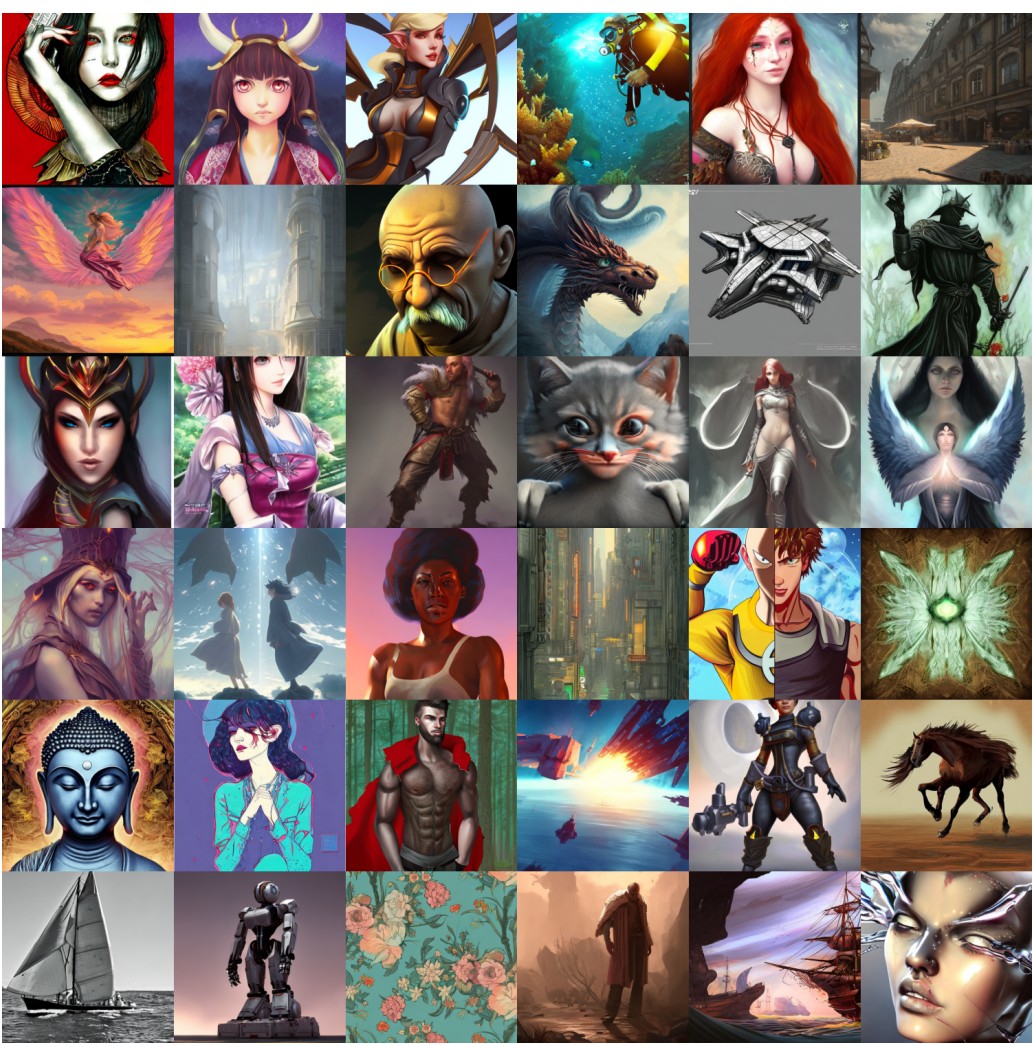

Figure 10: More watermarked images generated with WIND.

**Before**        **After**        **Before**        **After**

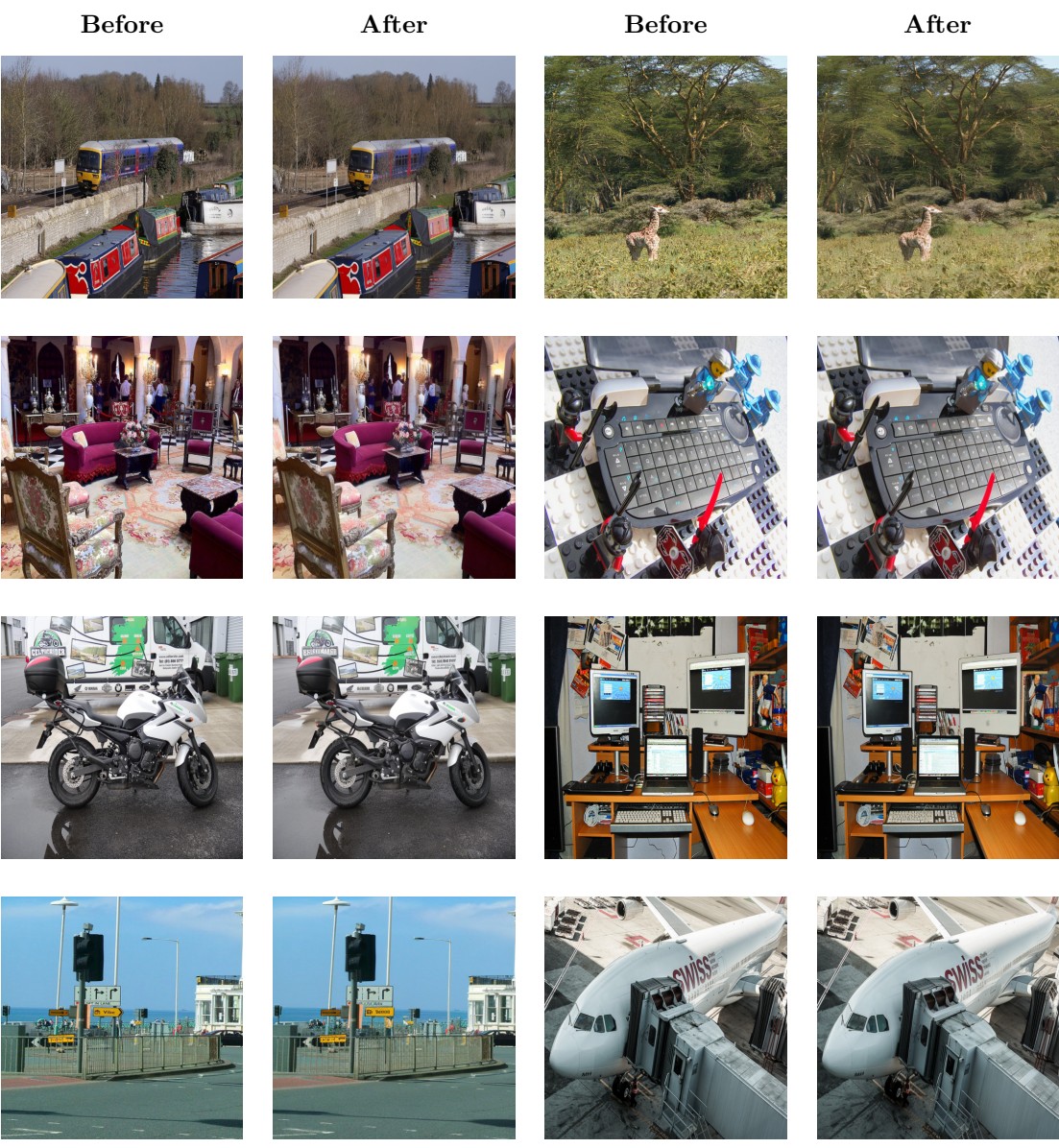

Figure 11: More comparisons of COCO images before and after watermarking with WIND.

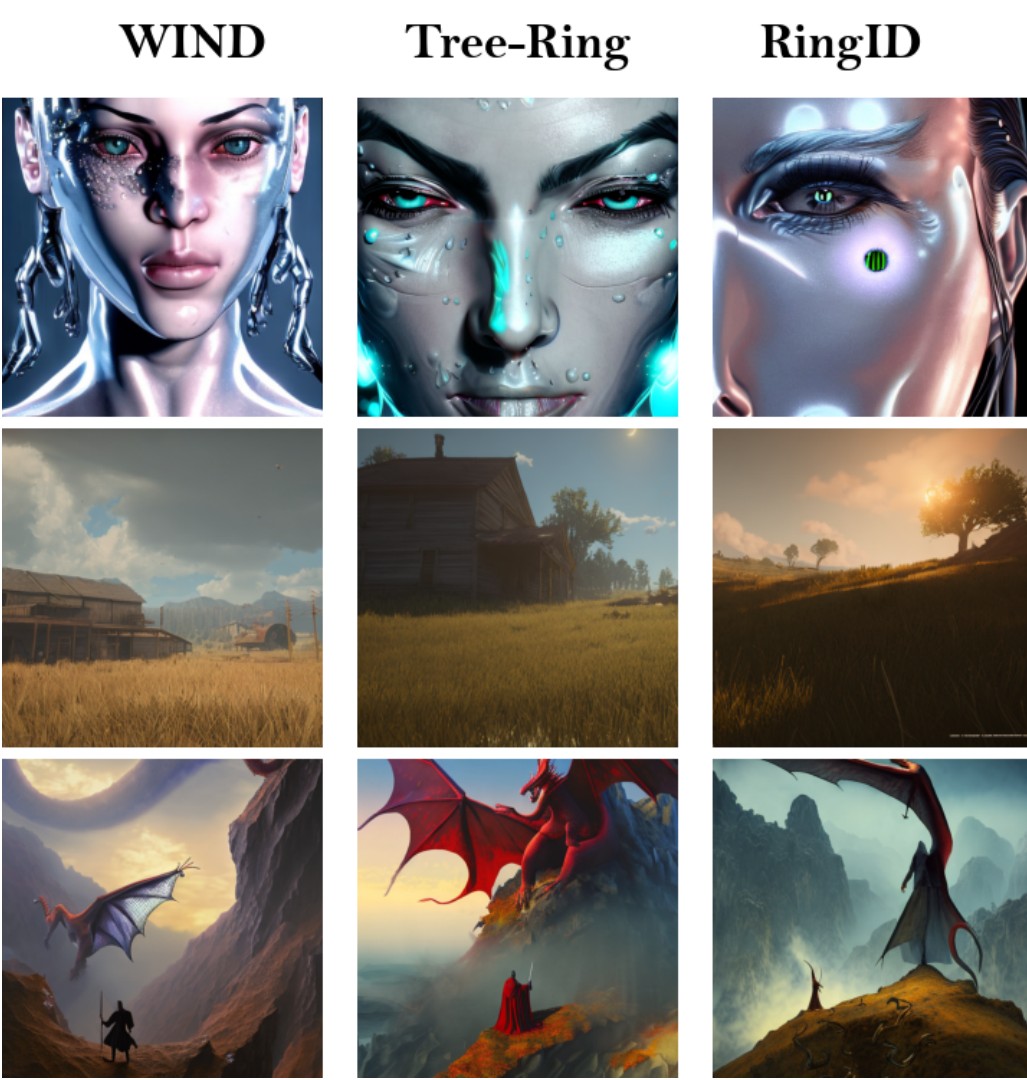

Figure 12: More qualitative results of watermarked images generated using WIND, Tree-Ring, and RingID.

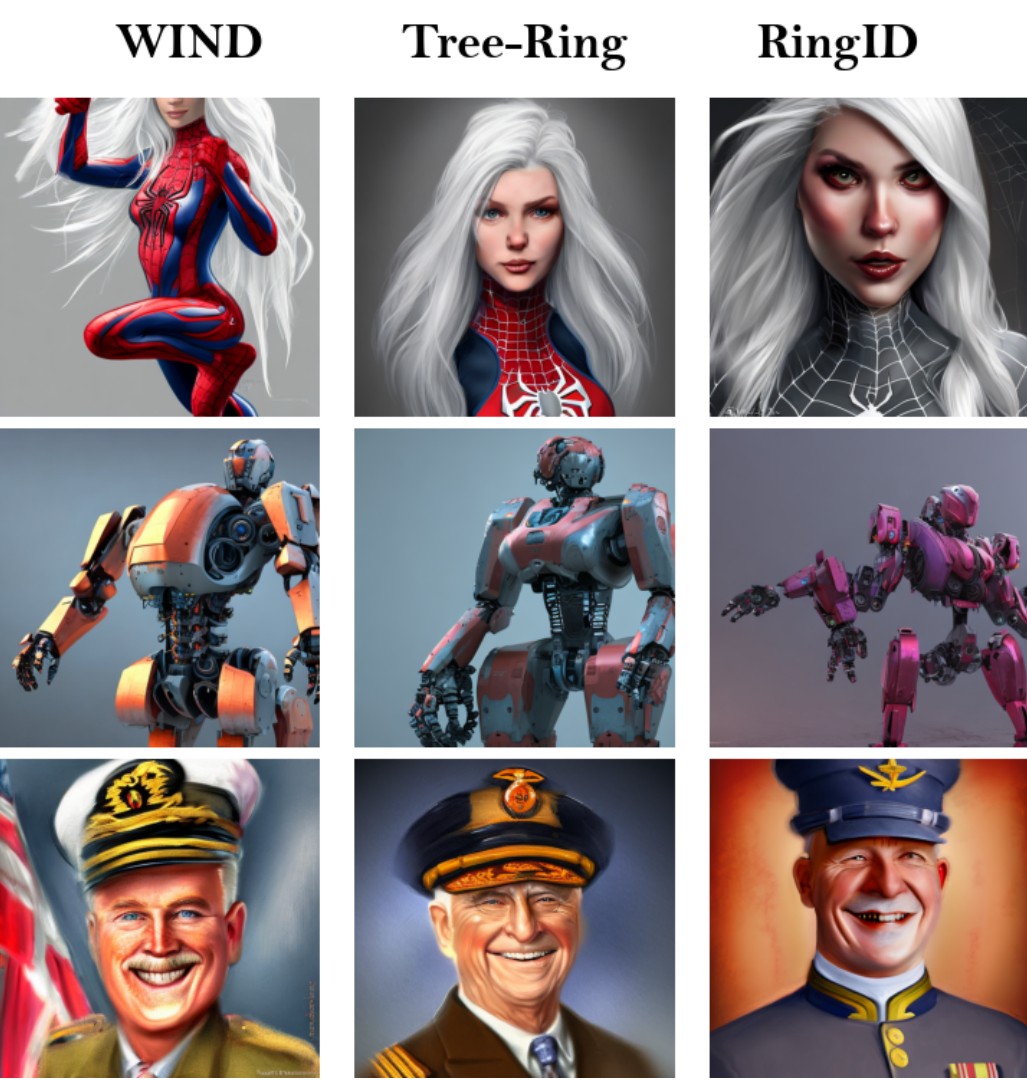

Figure 13: More qualitative results of watermarked images generated using WIND, Tree-Ring, and RingID.

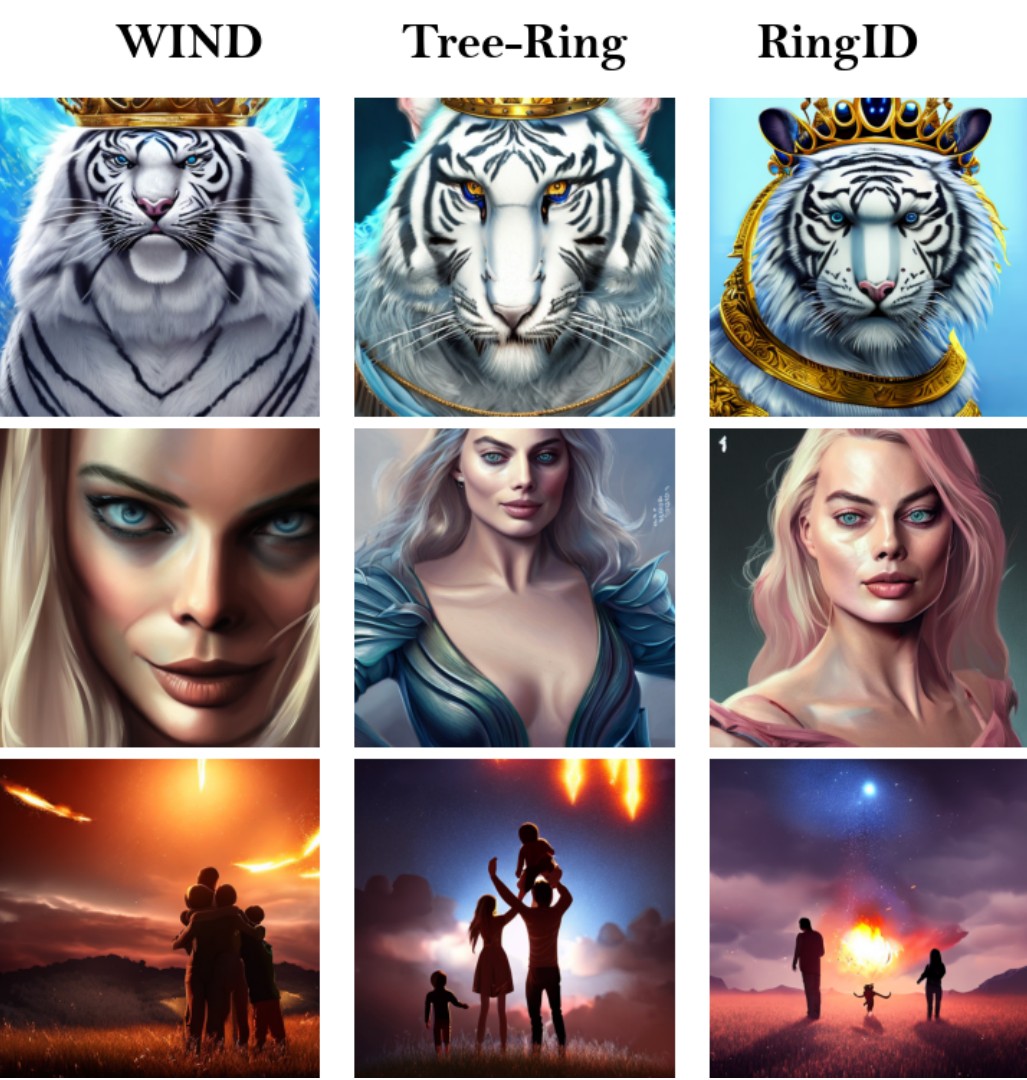

Figure 14: More qualitative results of watermarked images generated using WIND, Tree-Ring, and RingID.

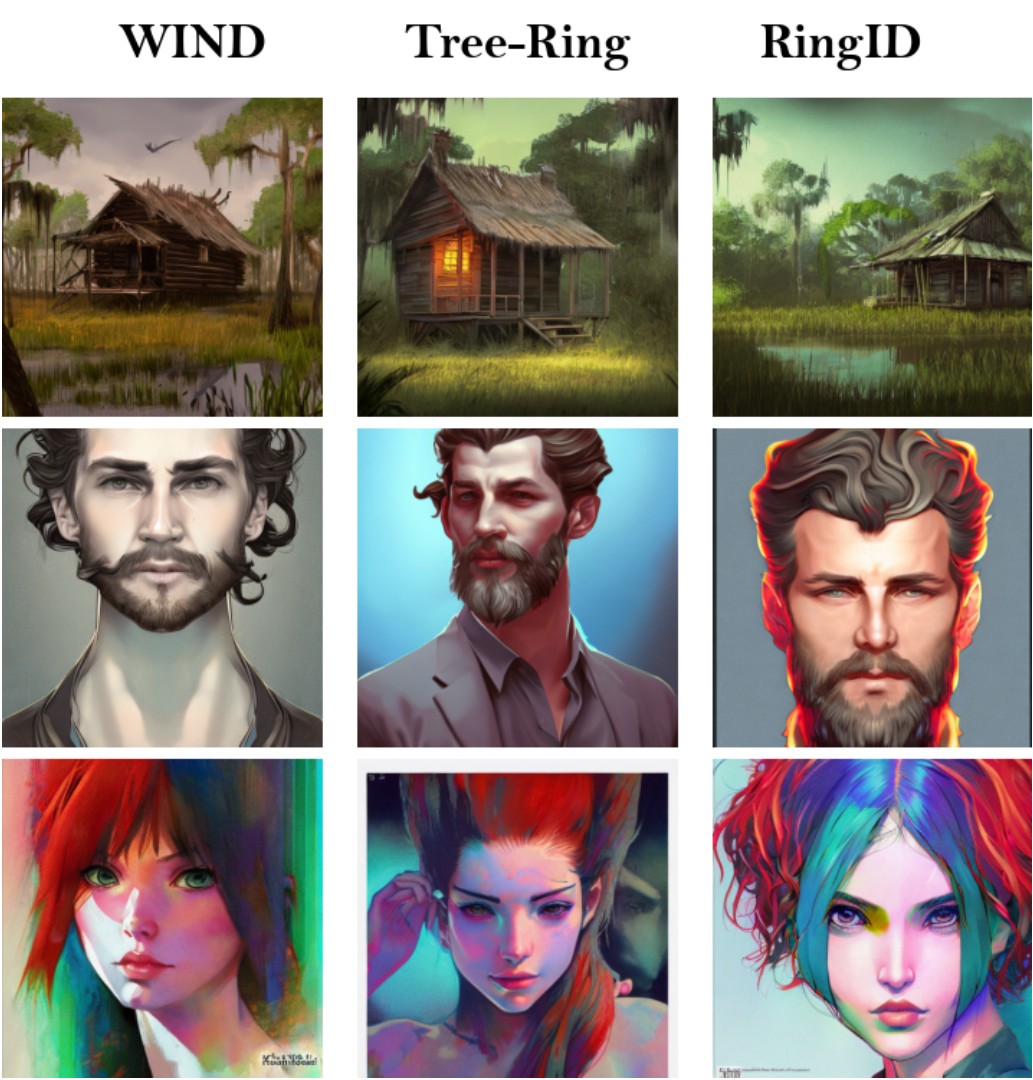

Figure 15: More qualitative results of watermarked images generated using WIND, Tree-Ring, and RingID.

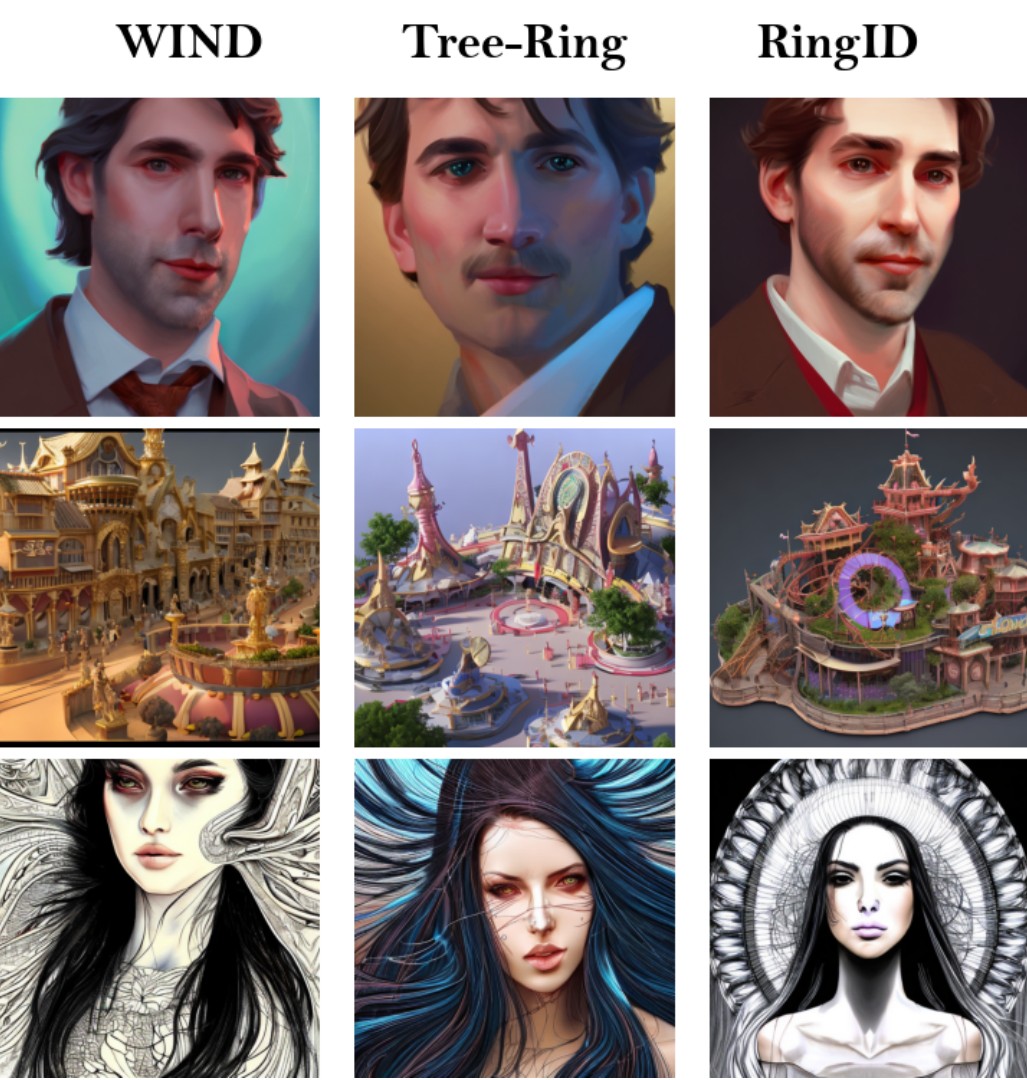

Figure 16: More qualitative results of watermarked images generated using WIND, Tree-Ring, and RingID.

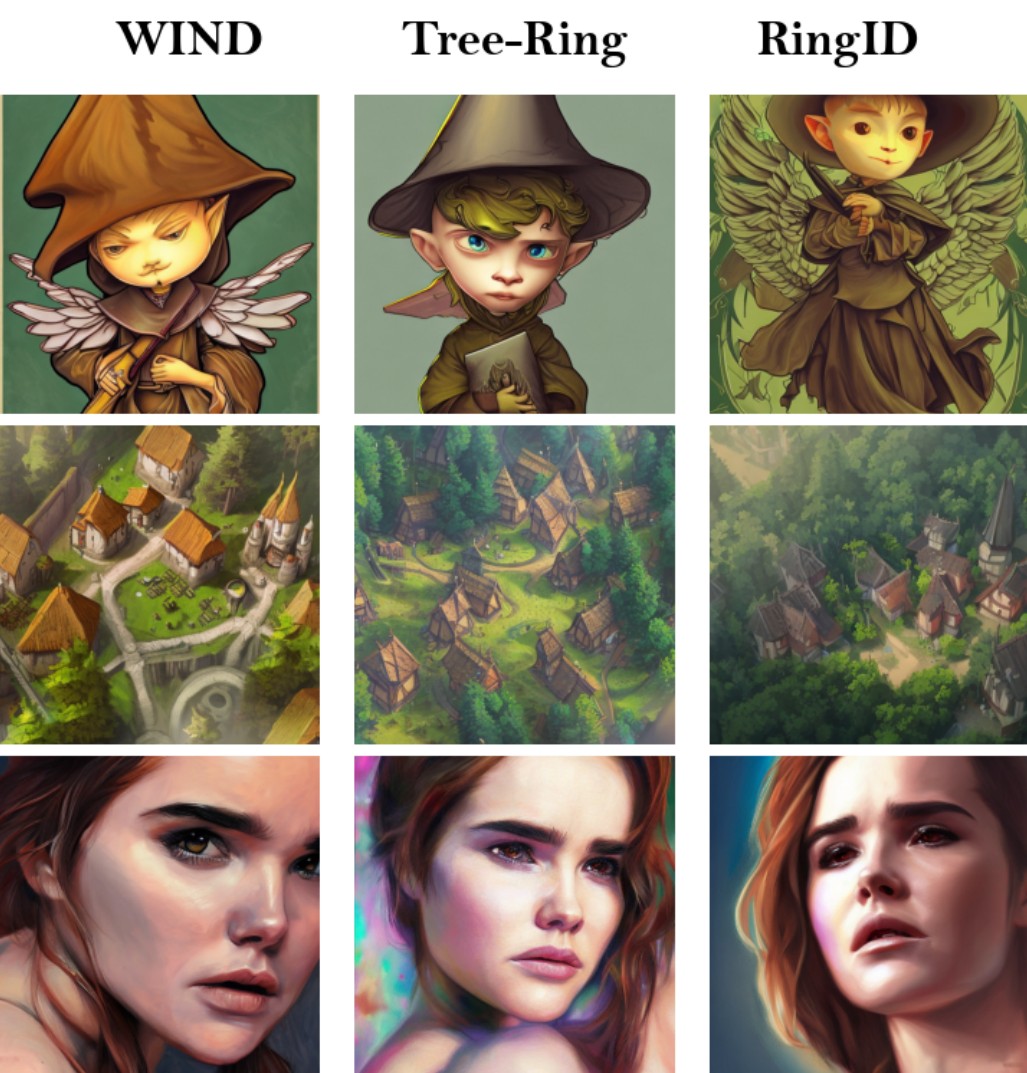

Figure 17: More qualitative results of watermarked images generated using WIND, Tree-Ring, and RingID.

