# OpenReview forum: "Hidden in the Noise: Two-Stage Robust Watermarking for Images"
_NeurIPS.cc/2024/Workshop/SafeGenAi — SafeGenAi Poster_

### Official Review · Reviewer_P3Rd · 2024-10-09
**The proposed approach aims to combat deepfakes while also enabling model owners to protect their intellectual property by introducing a novel watermarking method that is resilient to removal and forgery attacks. The authors compare their approach against existing methods such as Tree-Ring and RingID.**

**Rating:** 6
**Confidence:** 3

**Review:**

Strengths:

  1.  The concept of leveraging initial noise from the diffusion process as a watermark is innovative and compelling.
   2. The empirical evaluation demonstrates that WIND is highly effective in achieving robust watermarking, providing a solid foundation for its claims.

Weaknesses:

 1. It would be valuable to explore how this method could be extended to scenarios where a single instance of initial noise is used as a watermark, rather than a collection of noises.
 2. The approach, similar to other watermarking techniques, is heavily dependent on inversion, a limitation that the authors have acknowledged in the discussion.

---

### Official Review · Reviewer_MVXU · 2024-10-09
**Comments**

**Rating:** 9
**Confidence:** 3

**Review:**

Summary

The paper presents a novel two-stage watermarking method called WIND for images generated by diffusion models. It leverages the initial noise in the generation process as a distortion-free watermark, enhancing robustness against forgery and removal attacks. The method combines initial noise with Fourier patterns to create a group identifier, allowing for efficient detection while maintaining high accuracy. The authors empirically validate the approach against various attacks, demonstrating significant improvements in watermark robustness compared to existing methods.

Strengths

1. The WIND method shows state-of-the-art resistance to both removal and forgery attacks, addressing a critical challenge in image watermarking for AI-generated content.

2. By utilizing a two-stage approach that categorizes initial noises into groups, the method reduces the search space during detection, improving runtime without compromising accuracy.

3. The authors provide extensive experimental results that validate the effectiveness of their method across various transformation attacks, showcasing its practical applicability.


Weaknesses

1. The robustness of the watermarking technique relies heavily on the security of the private model. If the model is compromised, the watermarking may be rendered ineffective.

2. While the paper addresses several types of attacks, it may not account for all possible future attack vectors, leaving room for potential vulnerabilities as new techniques emerge.

Questions:

How does the WIND method specifically mitigate the vulnerabilities associated with existing image watermarking techniques [1], particularly against forgery attacks?

What are the practical implications of using a private model for watermarking, and how might this affect the accessibility of the technology for broader use cases?

[1] Tree-Rings Watermarks: Invisible Fingerprints for Diffusion Images. In NeurIPS 2023.

---

### Official Review · Reviewer_gwa3 · 2024-10-11
**Promising Results in image watermarking: Clear Accept**

**Rating:** 8
**Confidence:** 4

**Review:**

## Paper Summary:
This paper proposes an innovative two-stage image watermarking framework that enables easy detection. The authors claim that the initial noise embeds information about noise groups during generation, which can be retrieved during detection. The paper states that the approach achieves state-of-the-art robustness against forgery and attacks.
## Strengths:
- This paper presents innovative work testing the capability of LLMs to memorize encrypted data, with potential applications as a watermarking scheme or secret communication channel.
- It also outlines the pros and cons under various configurations.
## Weaknesses:
- Zooming in on parts of the figure would better highlight the details demonstrating the method's effectiveness.
- Additionally, testing whether the watermark embedding is preserved with deep learning-based super-resolution would be beneficial.
- From Table 1, there is a significant performance gap between fast mode and full mode. A trade-off setting between the two is recommended to provide more options for users.
## Justification of Rating:
The paper demonstrates good performance in image watermarking under various transformation attacks, with the proposed method being robust and distortion-free. However, as mentioned in the limitations section, some attack methods were not considered. The paper is well-organized with clear writing, and extensive experiments showcase its effectiveness. Overall, I rate this paper as Clear Accept.